# Vaginal Bipolar Radiofrequency Treatment of Mild SUI: A Pilot Retrospective Study

**DOI:** 10.3390/medicina58020181

**Published:** 2022-01-25

**Authors:** Paolo Mezzana, Ignacio Garibay, Irene Fusco

**Affiliations:** 1Plastic Surgery Department, Delle Medical Center, 00191 Rome, Italy; pmezzana@gmail.com; 2Private Practice Istituto Laser Y Luz Pulsada de Mexico, Guadalajara 44670, Jalisco, Mexico; ignacio.garibay@gmail.com; 3Department of Pharmacology, University of Florence, 50121 Florence, Italy

**Keywords:** SUI, bipolar RF, endovaginal treatment, anterior vaginal wall, neocollagenesis

## Abstract

*Background and Objectives*: This retrospective study investigates the action of a bipolar, temperature controlled, endovaginal RF handpiece for the treatment of mild, moderate, and severe stress urinary incontinence with a minimally invasive approach. Stress urinary incontinence (SUI) is a common condition resulting in involuntary urine leakage, with an associated social and psychological impact. SUI is the most common type of urinary incontinence in women. *Materials and Methods*: We retrospectively studied 54 patients for this study. The bipolar radiofrequency energy used in all patients was 50 W, with temperatures maintained between 41 °C and 44 °C. Two sessions were performed four weeks apart. In order to monitor all patients before the first treatment and 4 months after the second treatment, the International Consultation on Incontinence Questionnaire-Short Form (ICIQ-SF) was used. Paired Student’s *t* test was used to elaborate the statistical data. *Results*: The average frequency of urine leak improved from “2–3 times a week” (2.1 ± 1.3 points before the treatment) to “once a week” (0.8 ± 1.3 points 4 MFU post-treatment). The average volume improved from “small/moderate quantity” (3.2 ± 1.6 points before the treatment) to “none” (0.9 ± 1.4 points 4 MFU post-treatment). No adverse events or side effects were found. *Conclusion*: Our preliminary results represent a good starting point to check the effectiveness and validity of the bipolar radiofrequency temperature-controlled method in the treatment of SUI.

## 1. Introduction

Stress urinary incontinence (SUI) is an involuntary loss of urine due to an increase in intra-abdominal pressure: this can be caused by slight or vigorous movement or physical effort, such as laughing, coughing, sneezing, and running [1]. SUI has a strong negative impact on the psychological, social, relational, and health condition of patients [2]. The aetiology of SUI can depend on multiple factors, such as an insufficient support of the pelvic organs, change in the intrinsic urethral closure mechanism, or a suspension of the anterior vaginal wall [3,4]. Conservative options include pelvic floor exercises and biofeedback. If conservative therapy fails, surgical procedures such as periurethral bulking agents, midurethral slings, Burch colposuspension, and fascial slings can be used [5]. However, histological findings have shown a collagen reduction in urethral walls in the case of sphincter dysfunction and/or loss of urethral support. Radiofrequency therapy may therefore be a suitable option due to its effect on collagen [6]. Radiofrequency (RF) uses the resistance of the skin or mucosal tissue to transform the RF energy supplied to the tissue into thermal energy. The generated heat, according to the temperature reached in the tissue, causes a tightening of the existing collagen and stimulates fibroblasts activation, leading to neocollagenesis [7,8]. The use of non-ablative steps of radiofrequency energy has been investigated in the recent past with transvaginal or transurethral techniques, and was found to shrink and stabilize the endopelvic fascia [9]. Both approaches are costly, have high rates of associated adverse effects (please list these here with a reference), and require antibiotic prophylaxis and sedation to complete [10,11,12,13,14]. Different from monopolar RF technology, the bipolar RF system is more efficacious [15] and works through two electrodes usually very close each other (maximum distance few millimeters in vaginal RF probes). The electrical current passes from one electrode to the other one, and it has no possibility to go around in the body. It remains confined just between the electrodes. In this way, we are absolutely sure that the electrical current is confined just between the electrodes.

In this study, the authors present the preliminary results of a bipolar, temperature controlled, endovaginal RF handpiece minimally invasive treatment used to treat mild, moderate, and severe stress urinary incontinence with a minimally invasive approach.

## 2. Materials and Methods

### 2.1. Patient Selection

This study was conducted in 54 patients enrolled at Delle Medical Center, Rome, Italy (Plastic Surgery Department) between February 2019 and August 2020. Women’s age ranged from 40 to 71 years, with an average age of 50.7 years. Patients were all normal weight (with a body mass index < 25 kg/m^2^), and 77% were sexually active and 67% were parous.

### 2.2. Pre-Treatment Indications

A sufficient amount of clear water-based warm gel was used as a conductive medium and was applied to the head of the handpieces covering all the electrodes. Conservative parameters were initially used (low power and short exposure times) and were gradually increased where tolerated [16].

### 2.3. Study Protocol

The sterile and disposable RF Touch handpiece (DEKA M.E.L.A. Calenzano, Italy) was used for the study. Patients were treated with this device, which received CE marking in February 2019 for the treatment of SUI. This handpiece is equipped with a couple of electrodes each 2.28 cm long, and has an internal negative temperature Coefficient (NTC) temperature sensor. Two point five centimeter gradings are marked on the side of the handpiece to allow for assessment of the penetration depth while using the device. The handpiece is connected to a bipolar RF generator that works at 500 Hz frequency with a maximum power of 50 W. The ergonomic shape of the handpiece allows for both internal and external vulvo-vaginal treatments. The treatment is performed only in the anterior portion of the middle and distal thirds of the vagina. The treatment is performed with clockwise and anti-clockwise movements to cover the area from 11:00 a.m. to 01:00 a.m., both middle and distal, for 5 min per part for a total of 10 min of treatment (see Figure 1).

The user can select the time of treatment per selected area and the minimum and maximum value of temperature for the RF treatment within the 35 °C and 45 °C range with 1 °C steps. If, during treatment, the detected temperature overcomes the maximum set temperature, the system will block emission until the value returns within the correct limit. If the temperature is lower for the minimum temperature set, the counter stops the treatment countdown. In this way, each selected quadrant is properly treated. The bipolar RF energy used in all patients was 50 W, with temperatures maintained between 41 °C and 44 °C and a total count of 42 for each quadrant (5 min) [16]. These temperatures were chosen to trigger collagenogenesis without causing its damage for shrinkage. For each patient, two quadrants on the vaginal anterior wall were treated: the first quadrant is at 5 cm, and the second quadrant is at 2.5 cm. Two sessions were performed four weeks apart. Patients may feel a sensation of heat during the procedure and may regain normal activity on the same day. RF treatment is comfortable and possible for all patients as anesthesia is not required. The study was conducted according to the guidelines of the Declaration of Helsinki. Written informed consent was obtained from all participants (see Appendix B). In order to monitor all the patients before the first treatment and 4 months after the second treatment, the International Consultation on Incontinence Questionnaire–Short Form (ICIQ-SF) [17,18] was used, severity intervals for the ICIQ-SF: slight (1–5), moderate (6–12), and severe (13–21). Paired Student’s *t* test was used to test all of the outcome data for statistical significance with the SPSS program version 25.0 (IBM), where *p* levels < 0.05 were considered statistically significant. The status and severity of each patient’s stress urinary incontinence was assessed by a gynecologist. Exclusion criteria for the study were the following: cardiac implants/wearers of pacemakers, collagenopathies and skin pathologies, pregnancy and breast-feeding, infectious diseases, neoplasia or history of skin cancer, renal and hepatic deficiencies or dysfunctions, subjects with arrhythmia or any other severe known heart disorders, subjects with any implantable metal device and/or body piercing in the treatment area, subjects with diabetes or autoimmune disorders, subjects with coagulation disorders, patients with transplants, known sensitivity to the device, patients with an implanted deep brain stimulation systems, patients with haemorrhagic diatheses, history of keloid scarring or of abnormal wound healing, and patients with a degree of pelvic organ prolapse at stage II or greater, as well as several conditions and/or pathologies, which, according to their specific anatomical site, severity, and specific features pertaining to each case, could be a reason for excluding a patient wishing to undergo RF treatment. After a preliminary interview with the patient, the physician judged whether to perform, postpone, or exclude the treatment.

### 2.4. Post-Treatment Indications

Patients can return to normal activities just after treatment. In case of intravaginal treatment, we recommend stopping sexual intercourse 48 h after the RF session.

## 3. Results

The study was completed by all 54 patients, and according to the protocol, the treatment sittings were conducted. Outcome data and ICIQ-SF results are shown in Table 1 and Figure 2. The ICIQ-SF was used to assess the severity of SUI and grouped 7 (13%) patients as slight SUI, 33 (61%) patients as moderate SUI, and 14 (26%) as severe SUI. At the 4-month follow-up, an improvement in SUI symptomatology was shown with 35 (65%) patients experiencing slight SUI, 18 (33%) experiencing moderate SUI, and 1 (2%) experiencing severe SUI (Figure 3).

At the 4-month follow-up (4 MFU), the results were statistically significant (*p* < 0.0001). The average frequency of urine leak improved from “2–3 times a week” (2.1 ± 1.3 points before the treatment) to “once a week” (0.8 ± 1.3 points 4 MFU post-treatment). The average volume improved from a “small/moderate quantity” (3.2 ± 1.6 points before the treatment) to “none” (0.9 ± 1.4 points 4 MFU post-treatment). In addition, the average interference improved from 4.7 ± 2.4 points before the treatment, to 2.3 ± 1.9 points 4 MFU post-treatment. Raw data (parameters of ICIQ-SF: frequency, volume, and interference) at baseline and at 4 MFU after the last session are shown in Appendix A. No side effects or adverse events, such as infections (i.e., bacterial vaginosis, candida, folliculitis, genital herpes, urinary tract/vaginal infection, vulvovaginitis) or general genital disorders (i.e., vaginal discharge, coital bleeding, dysmenorrhea, dyspareunia, hypomenorrhea, vulvovaginal dryness, vulvovaginal erythema) were observed during the entire treatment period.

## 4. Discussions

Over the past decade, there has been an increase in minimally invasive devices for orgasmic dysfunction, vulvovaginal laxity and atrophy, and stress urinary incontinence. Non-invasive treatment is strongly recommended for SUI in women. Surgical procedures are more likely to be curative [19], but are more invasive and have more associated adverse events. On the other hand, the practice of physiotherapeutic exercises such as kegel exercises [20,21] have a reduced effectiveness because they are often not performed correctly and constantly over time by the patients (women often need to be motivated to routinely perform kegel exercises) [22]. Recently, the use of radiofrequency and lasers, such as Erb:Yag [23] or CO2 lasers [24,25], for the treatment of SUI and genitourinary syndrome of menopause (GSM), have shown promising treatment outcomes [26,27]. Evidence supports laser treatment as an alternative intervention for SUI [28]. However, because of the growing demand for non-invasive, non-ablative/coagulative techniques that do not require a patient hospitalization recovery time, radiofrequency treatment has been established in the vaginal rejuvenation field [29,30,31]. Furthermore, radiofrequency is a powerful tool for this treatment because of its ability to reach greater tissue depth in comparison to laser treatment. To date, literature data suggest that radiofrequency SUI treatment has a promising efficacy and carries a minimal risk of adverse events [32]. No major adverse events were reported in studies using transvaginal and transurethral applications of radio frequency energy. There were minor complications observed, with hematuria, urinary tract infection, urinary retention, hesitation, and dysuria being the most common. However, most of these adverse events are transient, and urinary tract infections can be treated with the simple administration of antibiotics [11,33]. This radiofrequency-based conservative treatment for SUI is not innovative, nevertheless the device offers temperature-controlled treatment time, which allows for reproducibility and total safety [34].

The RF effects have proven to be beneficial in mild to moderate SUI, as demonstrate in several studies [15,16,35,36].

In the medical field, radiofrequency is used to produce thermal energy (heat) generated by the resistance of the different layers of skin or mucosa to the passage of the radiofrequency current. Technically, the RF source generates an electric field which, once in contact with the skin, results in an oscillating electric current, which in turn induces the translational movement of charged atoms and molecules, hindering the rotation of the polar molecules. This “molecular movement” causes the local temperature to rise. This generated heat depends on the amount of delivered current and the exposure time. The energy is dispersed in a three-dimensional volume of fabric at controlled depths. The bipolar RF system is able to cause collagen contraction and boost the production of new collagen in dermal/mucosal structures, reaching temperatures between 40 °C to 45 °C on a tissue, which may stimulate fibroblasts to produce new collagen through the activation of heat shock proteins and through the initiation of the inflammatory cascade. When the generated temperatures get higher than 45 °C, some thermal injuries and pain are noticed at the skin level, while vaginal mucosa can tolerate higher temperatures, up to 47 °C, for instance, without visible thermal injury. The release of thermal energy that occurs when the RF electrode passes a current through the skin induces collagen denaturation and contraction, activation of fibroblasts, increased blood flow, neocollagenesis, and neovascularization, all of which aid in restoring the elasticity and moisture of the vaginal mucosa [37]. A possible involvement and activation of the sex steroid precursor dehydroepiandrosterone (DHEA), which supports vulvo-vaginal cell estrogen production, playing an important role in vaginal rejuvenating, is hypothesized [38]. The efficacy and safety of RF touch (DEKA M.E.L.A. Calenzano, Italy) has already been assessed in the study of Gonzalez Isaza *p* and Velez Rizo DL [36], which reported that 90% patients who met the criteria for SUI reported to have a 70% improvement in SUI, which was correlated with important changes in the scores of ICIQ-SF. In this study, patient condition improvement was reported at once after the last treatment, and it was further more significant after the 4-month follow-up examination. This amelioration is guided, over time, by the collagen remodeling process, which requires up to 90 days to complete, and which by favoring the urethral closing mechanism could thus explain why there was a better response after 4 months of follow-up. Therefore, the results obtained are comparable with the period of neo collagen production. The urinary loss clinical response was satisfactory for the patients investigated. This research demonstrated that the method is reliable, painless, and effective, without the risk of adverse effects. In terms of safety, the results were similar to those of the pilot study by Millheiser et al. [29], which shows that radiofrequency, used on vaginal introitus to treat vaginal laxity, reported a total improvement of 87% and no adverse effects. This type of result is predictable as radiofrequency promotes the phenomenon of angiogenesis/neocolagenesis, collagen contraction, and the growth factor infiltration, which reestablish vaginal mucosa elasticity [39]. Instead of this, in a systematic review on the intraurethral radiofrequency technique, a relative risk (RR) of 5.76 of pain/burning, 1.36 of a hyperactive detrusor, and 0.95 of urinary retention was found [14].

Further future studies with longer follow-ups will be needed to understand how to produce outcomes over time.

### Study Limitations

The limitations of the current study include small numbers in each group and a lack of long-term follow-up. Further studies with a longer follow-up are needed to confirm our findings. Our future goal is therefore to develop randomized clinical trials to evaluate the large-scale efficacy of the method.

## 5. Conclusions

Our preliminary results represent a good starting point to assess the effectiveness and validity of the bipolar radiofrequency temperature-controlled method in the treatment of SUI. All subjects tolerated treatment well, without showing any adverse effects.

## Figures and Tables

**Figure 1 medicina-58-00181-f001:**
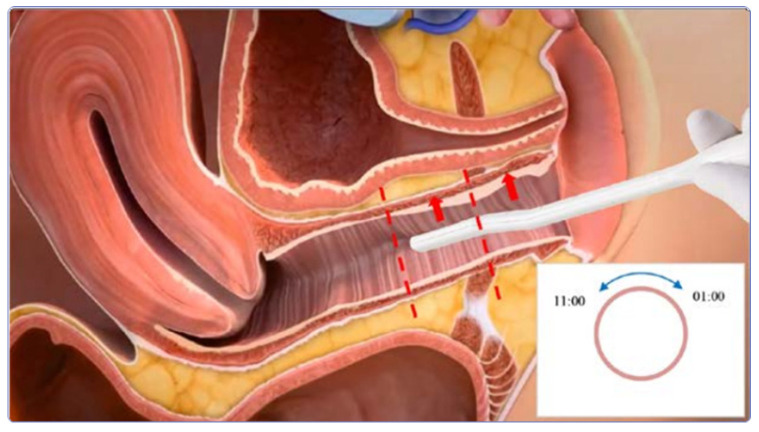
Schematic representation of a vagina cross-section and indications for handpiece use for SUI treatment procedure. Courtesy of DEKA M.E.L.A.

**Figure 2 medicina-58-00181-f002:**
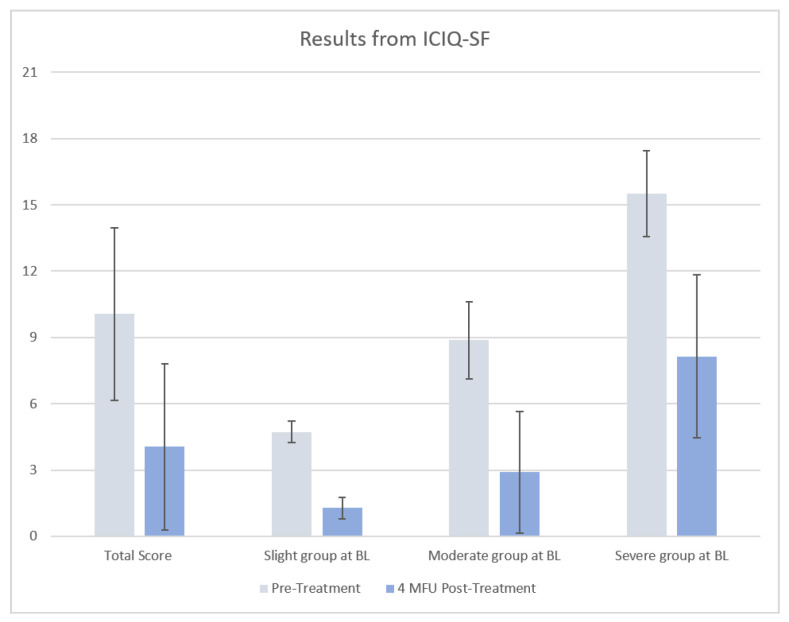
Results from ICIQ-SF at baseline and at 4 MFU: total score for all patients; variation of ICIQ-SF score for patients with slight severity at baseline (BL), with moderate severity at baseline (BL) and severe severity at baseline (BL).

**Figure 3 medicina-58-00181-f003:**
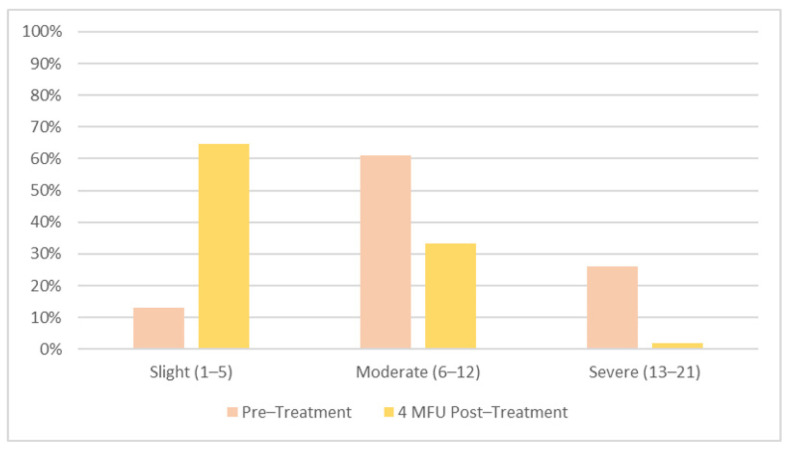
Patient incontinence severity at baseline and 4 MFU after the last session; correlation between the ICIQ-SF score and incontinence severity index: slight (range score: 1–5), moderate (range score: 6–12), and severe (range sore: 13–21).

**Table 1 medicina-58-00181-t001:** Results from ICIQ-SF.

Questionnaire ICIQ-SF	Score Range	Pre-Treatment	Post-Treatment	*p*-Value
Frequency	0–5	2.1 ± 1.3	0.8 ±1.3	*p* < 0.0001
Volume	0–6	3.2 ± 1.6	0.9 ± 1.4	*p* < 0.0001
Interference	1–10	4.7 ± 2.4	2.3 ± 1.9	*p* < 0.0001

## Data Availability

The data that support the findings of this study are available from the corresponding author upon reasonable request.

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
