# Peer review of "Vaginal Bipolar Radiofrequency Treatment of Mild SUI: A Pilot Retrospective Study"

_medicina, 2022, doi:10.3390/medicina58020181_

Round 1

Reviewer 1 Report

Vaginal Bipolar Radiofrequency Treatment of Mild SUI: a Pilot Retrospective Study.

Abstract

Change RF to radiofrequency

Rather than saying mild and high effort SUI- change to mild, moderate and severe SUI

Rather than saying: Stress urinary incontinence (SUI) is a common condition that consists in involuntary urine leakage that results in social problem and it is frequently associated to vaginal laxity- change to Stress urinary incontinence (SUI) is a common condition resulting in involuntary urine leakage, with an associated social and psychological impact  

Change to “used to evaluate”

Change to “tested using a paired student T test”

Intro

Line 38: Change to- Conservative options include: pelvic floor exercises and biofeedback.

Line 40: If conservative therapy fails, surgical procedures such as periurethral bulking agents, midurethral slings, Burch colposuspension and fascial slings can be used.

Line 44- change to …urethral support. Radiofrequency therapy may therefore be a suitable option due to its effect on collagen.

Line 48- change to: leading to neocollagenesis

Line 51- change to transvaginal or transurethral

Line 52- change to: Both approaches are costly, have high rates of associated adverse effects (please list these here with a reference), and require antibiotic prophylaxis and sedation to complete.

Line 56- Change to mild, moderate and severe

Materials and methods

Line 61- change to- Women’s age ranged from

Line 62- change to- Patients were all normal weight (with a body mass index < 25kg/m2). Seventy seven percent were sexually active and 67% were parous.

Line 66- please quantify the amount of RF conductive gel

Line 67- in women with a lower pain threshold, the gel was preheated for 1 minute of emission to reduce discomfort.

  • Please add a reference here to show this to be an effective method of pain reduction

Line 69: change to- Conservative parameters were initially used (low power and short exposure times) and  were gradually increased where tolerated

  • Please add a reference here to show this is standard procedure

Line 78- change to- Two point five centimetre gradings are marked on the side of the handpiece to allow for assessment of penetration depth while using the device

Line 82- the treatment is performed

Line 83- the treatment is performed with a clockwise…

Line 90- change to- … correct limit. If the temperature …

Line 93- Please add a reference here to show this is standard procedure

Line 94- chosen- Please add a reference here to show this is standard procedure

Line 100- please insert the REB study approval number

Line 101- please include this consent in the appendix

Line 107- why is the urogynaecologist

Questions about study methodology:

Why was there no control group?

Results

Line 131- change to : the ICIQ-SF was used to assess severity of SUI and grouped 7 (13%) patients as slight SUI, 33 (61%) patients as moderate SUI and 14 (26%) as severe SUI.

  • Please change to SUI throughout the text

Line 133- change to- at the 4-month follow -up an improvement in SUI symptomatology was shown with 35 (65%) patients experiencing slight SUI, 18 (33%) experiencing moderate SUI and 1 (2%) experiencing severe SUI (Figure 3).

  • Can you please do a chart showing the relative improvement in the different groups?

Line 138- can you stratify this into each group- slight, moderate, severe with the relative improvements in each group- perhaps a graph might illustrate this well.

Discussion:

Needs significant restructuring

Line 174: Please remove: SUI treatment options were limited to topical application, hormonal treatments, kegel exercises, and surgical options such as labiaplasty, vaginoplasty, and perineoplasty [17]- this is not correct

Line 178: Please change to: Surgical procedures are more likely to be curative (18) but are more invasive and have more associated adverse events

Line 184- please remove- Local estrogen therapy is used in the treatment of urge urinary incontinence, not SUI

Line189- please change to … promising treatment outcomes

Line 197-217- please reference

  • There is too much information here and this should be in the introduction not the discussion

Line 219-225- please move to introduction

Line 234- change compatible to comparable

Line 234- please remove: The temperature range control of the RF Touch handpiece is responsible of homogenous positive response to this technique in accurately selected patients. The goal is to heat the tissue to 41–44°C for a defined treatment timeframe

  • Please include discussion on potential side effects and compare to the results of this study
  • Please discuss the use of RF in severe vs mild SUI and compare with results of this study
  •  

Conclusion:

Line 255- change to: to assess the effectiveness

Reviewer 2 Report

The authors present a retrospective trial that described the use of transvaginal RF to treat SUI. The structure is correct, but some questions were raised while studying it.

Abstract:

The authors say: “International Consultation on Incontinence Questionnaire – Short Form (ICIQ-SF) was used to evaluated all the patients before the first treatment and 4 months after the second treatment.” They should write “to evaluate”.

Introduction:

The authors refer that slings and periurethral bulking agents are conservative treatments. Both treatments are surgeries and are performed in the operation theater under anesthesia. Please correct it.

Materials and Methods:

2.2. Pre-Treatment indications

The authors tell us that: “A good amount of RF conductive gel was applied to the head of the handpieces, covering all the electrodes, and to the labia majora and the other areas to be treated.” A “sufficient” amount fits better.

Finally, the authors wanted to present the usefulness of the transvaginal RF to women suffering from UI. They tell us that other methods are more expensive. Could they please inform us about the cost?  The trial presents an innovative technique.  

Round 2

Reviewer 1 Report

Vaginal Bipolar Radiofrequency Treatment of Mild SUI: a Pilot Retrospective Study.

Heading- mild, moderate and severe SUI (all three are included)

Abstract

Line 18 change to radiofrequency (RF)

Intro

Line 49- insert RF after radiofrequency (RF)- and use this abbreviation throughout the remainder of the text

Line 51- use RF and remove radiofrequency

Line 58- please list the adverse effects of RF and reference appropriately

Line 62- change differently to: In contrast to monopolar…

Line 65- change to: it is not possible for its energy to be dissipated throughout the body

Line 65: change to:  The electrical current remains confined between the electrodes. Remove the line- in this way we’re absolutely sure that the electrical current is confined just between the electrodes

Line 122- please insert the REB study approval number

This was not answered:

Questions about study methodology:

Why was there no control group?

Results

Figure 3: Please explain MFU ?months follow up

Discussion:

Major change required- this entire section needs to be re formatted- a lot of this information should be in the introduction. There is no reference to the study findings in the discussion- this needs to be included

As previously requested:

  • Please include discussion on potential side effects and compare to the results of this study
  • Please discuss the use of RF in severe vs mild SUI and compare with results of this study

Study limitations:

Please comment on the lack of a control group.
